# Falls and Preventive Practices among Institutionalized Older People

**DOI:** 10.3390/ijerph19137577

**Published:** 2022-06-21

**Authors:** Cristina Lavareda Baixinho, Carla Madeira, Silvia Alves, Maria Adriana Henriques, Maria dos Anjos Dixe

**Affiliations:** 1Nursing Research, Innovation and Development Centre of Lisbon (CIDNUR), 1900-160 Lisbon, Portugal; ahenriques@esel.pt; 2Vila Franca Hospital, 2600-009 Vila Franca de Xira, Portugal; carla.madeira@hvfx.pt (C.M.); silvia.alves@hvfx.pt (S.A.); 3Center for Innovative Care and Health Technology (ciTechcare), School of Health Sciences of Polytechnic of Leiria, Polytechnic of Leiria, 2410-541 Leiria, Portugal; maria.dixe@ipleiria.pt

**Keywords:** accidental falls, aged, institutionalization, nursing homes, practices, risk

## Abstract

The present study about falls among the older adult population essentially explores bio-physiological factors. In light of the complexity of the factors that cause these accidents, it is important to identify the safety and self-care practices of institutionalized older adults and their relationship with falls in order to introduce prevention measures and personalized cognitive–behavioral strategies. The objective of this study was to: (a) determine the frequency of falls and their recurrence among institutionalized older adults, and (b) to associate the occurrence of falls among institutionalized adults with or without cognitive impairment to communication and safety practices. This was a quantitative, correlational, and longitudinal study with 204 institutionalized older adults living in two long-term care facilities in Portugal. The Scale of Practices and Behaviors for Institutionalized Elderly to Prevent Falls was administered to the sample. The prevalence of falls at a 12-month follow-up was 41.6%, of which 38.3% were recurring episodes. Older adults with cognitive decline showed lower mean scores for safety practices. Further research with larger samples should explore the relationships between communication and safety practices and falls, their recurrence, and fear of new falls.

## 1. Introduction

Falls among the older adult population is a collective health problem because of their negative impact on the functionality of older adults [1,2,3,4,5]. These events can occur in all phases of human life but are especially more prevalent among the population of institutionalized older adults [1,2,3,4,5]. A study conducted in the Netherlands indicated an annual prevalence of 1471/1000 people, with 345/1000 suffering injuries (mild injuries—199/1000, moderate—93/1000, severe—37/1000, with 17 fractures of the femoral head per 1000 residents) [1]. Other studies have shown that 50.2% of residents suffered at least one fall in the previous year [2], with an average of 1.3 ±0.48 falls [5] per 1.57 ± 2.78 residents [2].

There is consensus that two or more risk factors contribute to falls, making it a multifactorial phenomenon, to which bio-physiological, behavioral, environmental, and socio-economic risk factors contribute [4,5,6]. A literature review indicated that the biomedical model has dominated research, given that most studies associate falls with bio-physiological risk factors [1,2,3,4,5,6], highlighting cognitive changes [4,5,6], male gender, visual changes [4], postural instability [5], changes in gait [6], polymedication, psychotropic therapy, independence in carrying out activities of daily living, previous falls [5], fatigue [3], and the use of selective serotonin reuptake inhibitors or serotonin and norepinephrine reuptake inhibitors [4]. Moreover, although the consumption of benzodiazepines is usually associated with the occurrence of falls, one study showed that their consumption appeared to be associated with a lower risk of falls [4], while another considered this consumption a risk factor, especially among older adults with cognitive decline, given that most of those who had suffered falls were on benzodiazepines (65.9%), compared to those without cognitive decline (32.2%) [6].

Few studies have associated environmental causes [7] and the safety practices/behaviors of older adults [8] with the risk, occurrence, recurrence, and injuries of falls. Regarding environmental risk factors, long-term care facilities (LTCF) for older adults may be responsible for more than 30% of falls (*p* = 0.0005) [7]. In relation to practices, the results have shown that older adults were less persistent in their choice of closed shoes with slip-resistant outsoles. Men gave more importance to safety practices and behaviors (*p* = 0.045) than women. The most dependent older adults adopted fewer prevention practices and behaviors. Older adults who used walking aids showed better communication practices (*p* = 0.019) and best practices and behaviors related to the accessibility of their physical environment (*p* = 0.012) [9]. However, studies have not associated these practices with the occurrence of falls and associated injuries.

Other factors that have also been pointed out as having the potential to hinder the management of this problem include rapid changes in functional and cognitive status; lack of familiarity of staff with the needs of and standards of care for these patients; the absence of policies that impact health care; workload–staffing ratios; lack of education among professionals and older adults about safety interventions; lack of transition care between hospitals and LTCF (in terms of the occurrence of falls and risk assessment) [10].

There is consensus regarding the need to assess the risk of falls [4,7] and provide recommendations to systematically assess older adults who have fallen [3,4] in order to decide on the personalized prevention measures that should be introduced [6] and also to increase the evidence about factors that contribute to falls specific to this context [4]. However, a Cochrane literature review pointed to a lack of sustained evidence supporting interventions in LTCF, affirming that there are still doubts about the effect of exercise on the rate of falls and that it may make little or no difference in relation to the risk of falls; therapeutic conciliation may make little or no difference on rate of falls or risk of falls, and vitamin D supplementation probably reduces the rate, but not the risk [11].

Although the results of a systematic review with meta-analysis of randomized controlled trials about the effectiveness of exercise for fall prevention in nursing home residents suggested that exercise did not play a role in preventing falls [8], some studies observed that exercise programs are effective in reducing the fall risk in older adults, and the recommendations are that integrated training (resistance training, core training, and balance training) with a duration of over 32 weeks and a frequency of more than five times a week is more effective in reducing the fall risk in older adults [12]. A randomized control trial that aimed to determine the effectiveness of a multi-system physical exercise for fall prevention and health-related quality of life in pre-frail older adults concluded that the program significantly increased muscle strength and improved proprioception, reaction time, and postural sway leading to falling risk reduction in older adults with pre-frailty [13].

The urgency of prevention is justified, not only by the negative impacts that these accidents have on the quality of life of institutionalized older adults but also because every Euro invested in prevention saves EUR 2.68, of which EUR 2.31 go to LTCF in increased hours of nursing care related to treating injuries [1]. The authors also established a relationship between the number of hours of prevention vs. treatment of one to four, i.e., on average, for each hour of nursing spent on interventions, more than four hours of fall-related care are saved. Even in medical and paramedical care, each hour of work spent on intervention activities was compensated by 2.2 and 1.5 h saved, respectively, with treatment.

Given the above, the objectives of this study were: (a) to determine the frequency of falls and their recurrence among institutionalized older adults, and (b) to associate the occurrence of falls among institutionalized older adults with and without cognitive decline with communication and safety practices.

## 2. Materials and Methods

### 2.1. Study Design

This was a quantitative, correlational, and longitudinal [14] study. The older adults were assessed twice, with a 12-month interval between the administration of the instruments and fall assessments.

### 2.2. Participants

The participants were older adults in two Portuguese LTCF who met the previously defined eligibility criteria: age ≥ 65 years, and permanent residency in the LTCF. Older adults who came to the LTCF during the day but did not reside in the facility, and those who only underwent the first assessment (withdrawal from the study, death, change of facility, or hospitalization at the time of the second assessment) were excluded.

### 2.3. Data Collection

The data were collected between January 2019 and February 2020. The instruments were administered by a nurse in each facility who was trained by one of the researchers on how to administer them, and how to assess and record falls. They were also instructed on the objectives and methodological procedures of data collection.

Falls were defined according to WHO (code e880-E888 of International Classification of Disease-9) as any event that results in a person coming to rest inadvertently on the ground or floor or other lower level, excluding intentional change in position to rest on furniture, walls, or other objects [15]. After a fall episode, the professionals documented it as part of the older adults’ clinical process on a form created for this study.

The Mini-Mental State Examination (MMSE), Portuguese version [16] was used to assess mental state (≤15 points for illiterate patients, ≤22 for patients with up to 11 years of schooling, and ≤27 for patients with more than 11 years of schooling).

To determine the practices and behaviors adopted by the institutionalized older adults to prevent falls, the Scale of Practices and Behaviors for Institutionalized Elderly to Prevent Falls (EPICIPQ as per its acronym in Portuguese) [9,17] was used, which has two dimensions. The first measures bilateral communication practices between older adults and different team professionals, and the other dimension concerns the safety practices and behaviors adopted by the older adults (composed of two factors: safety behaviors regarding self-care, and accessibility of physical spaces). Each of the items is scored on a Likert scale between 1 (never) and 5 (always). The validation study showed that both dimensions had good internal consistency: the 6 items of the communication subscale presented an α = 0.881, and the safety practices and behaviors dimensions, α = 0.817, for 11 items [9,17].

The interview was carried out with older adults with no cognitive decline [9]. For those who presented cognitive decline, the scale was administered based on observation by the nurses who had information for the application of the instruments, in each facility. This instrument was validated to be used as both an interview script and an observation scale [17].

### 2.4. Data Analysis

The data were processed using the Statistical Package for the Social Sciences (IBM SPSS, Atlanta, GA, USA), version 23.0. Descriptive statistics were used: absolute frequencies, measures of central tendency (mean), measures of dispersion and variability (standard deviation). Inferential statistical techniques were used to study the relationship between the variables: the Student’s *t*-test (when the distribution of the variables approximated a normal distribution), and the Mann–Whitney U test (when the variables did not approximate a normal distribution) [14]. Crammer’s V and Cohen’s d effect size indices were also calculated, considering the following cut-off points: 0 to 0.19, insignificant; 0.20 to 0.49, small; 0.50 to 0.79, medium; 0.80 and above, high.

### 2.5. Ethical Considerations

This study was part of a broader project about fall risk management in LTCF and was approved by the Research Ethics Committee of the Institute Polytechnic of Leiria (RESOLUTION NO. CE/IPLEIRIA/46/2020). The authors abided by the ethical principles related to consent, privacy, and confidentiality in research with human subjects.

## 3. Results

At the time of the first assessment, 231 elderly people met the eligibility criteria. At the 12-month follow-up assessment, due to hospitalization or death, the sample fell to 204 older adults from the two LTCF, half of whom presented cognitive decline (73.5% female and 26.5% male); they had resided in the facility for an average of 43.29 ± 41.64 months. Regarding age, 2.9% were ≥65 <75 years old; 27.5% ≥75 <85 years old; and 69.6% were 85 years old or older.

At the 12-month follow-up, 41.6% of the older adults had suffered at least one fall, and 38.3% of the episodes were recurrent.

Regarding the practices and behaviors of bilateral communication between the older adults and health professionals and the safety practices and behaviors adopted by the older adults (associated with safe self-care behaviors and accessibility of physical space), the data presented in Table 1 show that there were differences between the older adults with or without cognitive decline.

The effect size was calculated for variables with significant results. In the communication practices and behaviors domain, the effect size was revealed to be high (1.5 with a 95% confidence interval). In the safety practices and behaviors domain, the registered effect size was high (3.34 with a 95% confidence interval).

Table 2 shows that although the practices and behaviors adopted to manage the risk of falls differed according to the older adults’ cognitive status, they did not differ according to whether the older adult had fallen within the same sample of older adults with or without cognitive decline, with the exception of older adults with cognitive decline who had fallen within that year. These individuals presented lower average values for communication and safety practices.

It should be noted that in both groups, on average, safety practices and behaviors were adopted at lower levels than communication practices.

Additionally, no gender-related differences were observed in the adoption of safety practices among older adults with and without decline (Table 3).

## 4. Discussion

It is necessary to increase knowledge about the behavioral factors that constitute risk or cause of falls, especially because falls and fear of falling also interfere with functional capacity, increasing dependence and the need for replacement in self-care while resulting in restrictions and limitations on freedom [17,18,19]. However, fall behaviors among older adults, which include both fall risk and fall prevention behaviors [20], have not been as extensively explored [20,21] as other fall risk factors such as medication, gait, balance, weight, and fear of falling, among others [5,6,7,8,9,10,11].

In a qualitative study [22] conducted with older adults on their return home after hospitalization, six themes associated with fall-related behaviors emerged: sedentary lifestyle and limited functionality; prioritizing social involvement; low perception of fall risk; attribution of risk to external factors; prevention and care as fall prevention; little information about fall prevention during the transition from hospital to home. Limited awareness and involvement in effective fall prevention can increase the risk of falls in recently hospitalized older adults. Prevention programs tailored to the post-discharge period can involve patients in fall prevention, promote well-being and independence, and connect hospital and community efforts [22].

A study conducted on community-dwelling older adults that used linear regression analysis associated more effective fall behaviors (R^2^ = 0.256) with being male [95% CI: 2.178 to 7.789, *p* < 0.001], having lower body mass index [95% CI:—0.692 to −0.135, *p* < 0.05], living with family [95% CI: 0.022 to 5.953, *p* < 0.05], and having greater functional mobility [95% CI:—2.008 to−0.164, *p* < 0.05] [23].

Another study carried out in an LTCF, which also used the Scale of Practices and Behaviors for Institutionalized Elderly to Prevent Falls, concluded that the older adults did not persevere in the adoption of safety behaviors and that men gave more importance to safety practices and behaviors (*p* = 0.045) than women [17]. The most dependent older adults presented worse prevention practices and behaviors [17]. The best communication practices (*p* = 0.019) and best practices and behaviors related to the accessibility of their physical environment (*p* = 0.012) [17] were adopted at higher levels by older adults who used walking aids. Despite these results, the authors did not associate practices with the prevalence of falls and their recurrence and with cognitive decline. The results of the present study showed that the occurrence of falls among persons with cognitive decline at the 12-month follow-up was associated with lower levels of safety and communication practices. Future studies on falling and its recurrence should associate falling with the safety practices and behaviors of the older person, with and without decline, to prove or disprove this result.

The results of this study also indicated that older adults with cognitive decline adopted better practices to prevent the recurrence of falls. This result may be influenced by the fact that formal caregivers feel greater responsibility for the safety of people with cognitive decline. A study that aimed to understand how the fear of falling arises and manifests itself in the caregivers of institutionalized older adults observed that in terms of preserving the safety of the older adults, both caregivers and health professionals expressed greater concern about the safety of those who were afraid of falling, and those who were frailer, more dependent, or presented cognitive decline [24]. This finding supports the need to strengthen the guidelines and support provided to carry out activities of daily living.

Risk behaviors, such as hastiness, carelessness, improper use of walking aids, wearing inappropriate shoes, and lack of exercise, can aggravate the risk of falls among older adults [20]. These behaviors appear to be influenced by the roles that older adults assign to their independence in making risk-related decisions; their experience with previous falls; their level of understanding of individual risks; their ability and willingness to have support; the need vs. the desire to conceal a history of falls, and the influence of socioeconomic factors on risk management [21].

Communication is a central element of fall prevention [11,25,26,27,28,29]. One study that evaluated a fall prevention program in an LTCF observed a decrease in the rate of falls of 36.3% after the introduction of a fall risk/intervention instrument and after the creation of communication opportunities that broadened and deepened data collection about the risk and ensured that fall risk management actions were widely shared, understood and applied consistently among older adults and professionals [26]. The results of the present study showed that communication practices among older adults and between them and caregivers were, on average, superior to safety practices. Future studies should explore the relationship between communication practices and the adoption of safety behaviors in terms of self-care and controlling environmental risk factors.

The level of concern/awareness of the risk of accidents can be an important factor for prevention, especially when older adults associate this concern with preventive measures when performing self-care, introducing changes in the environment to increase safety, and maintaining health surveillance [11,28,29]. We agree that in LTCF, safety regulations often predominate over any other consideration and become a source of suffering and oppression, conditioning individuals’ decisions regarding how to carry out their activities of daily living [19].

### Study Limitations

The choice of a convenience sample and its size limits the generalizability of the results. The characteristics of Portuguese LTCF, their organization, and different models of care delivery limit these data to their context. Future studies with large, randomized samples are necessary to associate the practices of older adults with the risk and prevalence of falls, in addition to fall-induced injuries.

## 5. Conclusions

The prevalence of falls was 41.6%, and 38.3% of these events were recurrent. Fall risk management practices and behaviors differed depending on the older adults’ cognitive status, but they did not differ depending on whether or not the older adult had suffered a fall within the same sample of older adults with or without cognitive decline, with the exception of those with cognitive decline who had fallen by the time of the 12-month follow-up assessment, who presented lower average values for safety practices.

Future studies should explore the different communication practices associated with the adoption of safety behaviors, in addition to communication and safety practices associated with the fear of (new) falls.

## Figures and Tables

**Table 1 ijerph-19-07577-t001:** Results of Student’s *t*-test relative to the practices and behaviors adopted by older adults to manage the risk of falls according to the cognitive status of older adults. Lisbon; 2020.

	Cognitive Status	N	Mean	SD	t	*p*
Communication practices and behaviors	No decline	100	15.79	1.91	−6.776	0.000
	Decline	102	17.30	1.18		
Safety practices and behaviors	No decline	100	26.23	4.28	−10.775	0.000
	Decline	102	31.45	2.34		

**Table 2 ijerph-19-07577-t002:** Results of Student’s *t*-test relative to the practices and behaviors of older adults to manage risk of falls according to older adults who had fallen at the first assessment and at 12-month follow-up by cognitive status. Lisbon; 2020.

Cognitive Status	Practices	Variables		N	Mean	SD	t	*p*
No decline	Communication practices and behaviors	Falls at first assessment	No	57	26.84	4.03	1.661	0.100
			Yes	43	25.41	4.50		
	Safety practices and behaviors		No	57	16.26	1.87	2.956	0.004
			Yes	43	15.16	1.79		
Decline	Communication practices and behaviors	Falls at first assessment	No	61	31.75	2.29	1.589	0.115
			Yes	41	30.97	2.61		
	Safety practices and behaviors		No	61	17.06	1.56	1.156	0.250
			Yes	41	16.68	1.73		
No decline	Communication practices and behaviors	Falls at 12-month follow-up	No	60	26.31	4.20	0.247	0.806
			Yes	40	26.10	4.43		
	Safety practices and behaviors		No	60	15.98	1.97	1.241	0.808
			Yes	40	15.50	1.79		
Decline	Communication practices and behaviors	Falls at 12-month follow-up	No	56	31.91	2.19	2.181	0.032
			Yes	46	30.86	2.629		
	Safety practices and behaviors		No	56	17.21	1.38	2.087	0.039
			Yes	46	16.54	1.85		

**Table 3 ijerph-19-07577-t003:** Results of the Mann–Whitney U test relative to the practices and behaviors adopted by older adults to manage the risk of falls, by sex. Lisbon; 2020.

Cognitive Status	Practices	Variables		N	Minimum	Maximum	Median	Mean Rank	U	*p*
No decline	CPB ^(a)^	Sex	Male	30	15	18	18	56.42	872.500	0.170
			Female	70	14	18	18	47.96		
	SPB ^(b)^	Sex	Male	30	24	33	33	44.05	856.500	0.144
			Female	70	25	33	33	53.26		
Decline	CPB ^(a)^	Sex	Male	27	12	18	16.5	54.83	922.500	0.390
			Female	75	10	18	16	50.30		
	SPB ^(b)^	Sex	Male	27	17	33	25	57.63	847.000	0.147
			female	75	16	33	26.6			

^(a)^ Communication practices and behaviors; ^(b)^ safety practices and behaviors.

## Data Availability

Data are available only upon request to the authors.

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
