# Peer review of "Falls and Preventive Practices among Institutionalized Older People"

_ijerph, 2022, doi:10.3390/ijerph19137577_

Round 1
Reviewer 1 Report
Article is clearly presented aligning to the topic, although this is a topic that has been previously discussed in literature. I am aware that the authors have done this study in the context of Portugal, where they are based. The context and methodology of this study is well-presented.
Statistical calculations and presentation of results were appropriate and clearly presented. The authors have considered most if not all of the environmental and intrinsic factors which affect an institutionalized older person's risks of falls.
The discussion acknowledged the present literature available surrounding this topic and the authors appropriately discussed the controversies regarding certain preventive practices that have been suggested. Suggestions for future research directions in this topic is reasonable and well communicated in the conclusion.
From my review, I do not have further specific review suggestions to make.
Author Response
Dear reviewer:
Thank you for taking the time to review our article. Based on the recommendations of the reviewers we introduced the suggested changes and shaded it in yellow.
Reviewer 2 Report
Congratulations for your work, i add some questions and recommendations to improve the manuscript.
INTRODUCTION
You should add information about relationship physical activity and risk of falls as, TRX, BALANCE, STRENGTH, MULTICOMPONENTE EXERCISE PROGRAM.
METHODS
You should organize as sections (independent, dependent variables, study and design, statistical analysis)
RESULTS
Have you adjusted by basic confounder in the statistical analysis? sex, age, BMI, level education...
DISCUSSION
You should improve the conclussions, don't write number and %, you improve this part with POTENTIAL CONCLUSION OF THIS WORK.
Author Response

(The authors gave the same response as above.)

Reviewer 3 Report
Dear authors,
I have several concerns about the manuscript, especially in the introduction section. Several statements are out to date, and many statements must be explained to provide context.
Confidence intervals are required when there is any chance of type I/II error. As the means were too close in some results, please report the 95% CI of all p-values.
Some sentences lack references. Please, correct throughout the overall text.
Several recommendations were noted across the manuscript concerning further research. Focus on the present study would enable a clearer interpretation and facilitate readiness.
More comments can be found in the attached file.

Author Response

(The authors gave the same response as above.)

Round 2
Reviewer 2 Report
It's correct for to publish this manuscript. Congratulations.
Author Response
Dear review:
Once again we thank you for taking the time to review our article.
Reviewer 3 Report
Dear authors,
The manuscript has been improved, but many comments in the attached file were not corrected nor justified. Also, the mentioned statement must be updated and considered with other high quality studies that put exercises as key factors to prevent frailty and falls.
My recommendation remains the same.
Regards.
Author Response
Dear review:
Once again we thank you for taking the time to review our article.
We shade in yellow the change made in the introduction regarding physical exercise and its impact on preventing falls in nursing homes. We have shaded in green the changes made in the 1st revision based on the comments and suggestions you made on the pdf of the article, which you kindly made available to us.